# Switching to Brolucizumab in Neovascular Age-Related Macular Degeneration Incompletely Responsive to Ranibizumab or Aflibercept: Real-Life 6 Month Outcomes

**DOI:** 10.3390/jcm10122666

**Published:** 2021-06-17

**Authors:** Christof Haensli, Isabel B. Pfister, Justus G. Garweg

**Affiliations:** 1Berner Augenklinik am Lindenhofspital, 3012 Bern, Switzerland; isabel.pfister@augenklinik-bern.ch (I.B.P.); justus.garweg@augenklinik-bern.ch (J.G.G.); 2Department of Ophthalmology, Inselspital, University of Bern, 3012 Bern, Switzerland

**Keywords:** brolucizumab, ranibizumab, aflibercept, anti-VEGF, neovascular age-related macular degeneration, visual acuity, reading acuity, disease activity, treatment change, treat and extend

## Abstract

*Purpose*: The aim of this study was to evaluate the effect of switching treatment in eyes with neovascular age-related macular degeneration (nAMD) and treatment intervals of ≤6 weeks to brolucizumab. *Methods*: In this prospective series, eyes with persisting retinal fluid under aflibercept or ranibizumab every 4–6 weeks were switched to brolucizumab. Visual acuity (BCVA), reading acuity (RA), treatment intervals, central subfield thickness (CST), and the presence of intra- and subretinal fluid were recorded over 6 months. *Results*: Seven of 12 eyes completed the 6 month follow-up and received 4.4 ± 0.5 brolucizumab injections within 28.0 ± 2.8 weeks. Treatment intervals increased from 5.3 ± 0.9 weeks to 9.0 ± 2.8 weeks (95% confidence interval of extension (CI): 1.6 to 5.9). BCVA improved from 67.8 ± 7.2 to 72.2 ± 7.5 (95% CI: −0.3 to 9.1) ETDRS letters, RA improved from 0.48 ± 0.15 to 0.31 ± 0.17 LogRAD (95% CI: 0.03 to 0.25), and CST improved from 422.1 ± 97.3 to 353.6 ± 100.9 µm (95% CI: −19.9 to 157.1). Treatment was terminated early in five eyes (two intraocular inflammations with vascular occlusion without vision loss, one stroke, and two changes in the treatment plan). *Conclusions*: Improvement in visual performance and longer treatment intervals in our series over 6 months indicate the potential of brolucizumab to reduce the treatment burden in nAMD, while two instances of intraocular inflammation were encountered.

## 1. Introduction

The prevalence of neovascular age-related macular degeneration (nAMD) is increasing, along with an increasing treatment burden for patients and healthcare systems [1]. Current regulatory approved anti-vascular endothelial growth factor (anti-VEGF) drugs include ranibizumab since 2006 (Lucentis^®^, Novartis AG, Switzerland), aflibercept since 2012 (Eylea^®^, Bayer Pharmaceutical AG, Germany), and brolucizumab since 2020 (Beovu^®^, Novartis AG, Switzerland). Despite these options, disease stability is not achieved in all eyes, with averages of 62.9% and 56.0%, respectively, after 12 and 24 months of treatment, whereas treatment intervals may be extended during the same periods to ≥12 weeks, with stability at 37.7% and 42.6%, respectively [2]. HAWK and HARRIER, two phase 3 trials of brolucizumab, reported a better morphological stability of central retinal subfield thickness with noninferior visual acuity, compared to aflibercept [3]. ALTAIR, a phase 4 trials of aflibercept, investigated a treatment interval extension to 16 weeks under a treat-and-extend (T&E) protocol. In this trial, 43.9% of eyes maintained an interval of 16 weeks after 96 weeks [4]. Obviously, an unmet medical need exists with regard to the portion of eyes achieving disease stability and safely reaching treatment intervals of 3 months or more. Currently, the available limited evidence indicates an increasing potential to achieve disease stability and allow a treatment interval extension to ≥12 weeks from ranibizumab over aflibercept to brolucizumab [2]. The hope that brolucizumab might more effectively achieve disease stability in the subset of eyes which cannot be extended to 8 or more weeks under their current anti-VEGF therapy led to switching to brolucizumab with the aim of reducing the treatment burden in many cases upon its approval. Recent studies showed a short-term reduction of foveal thickness 4–8 weeks after switching to brolucizumab in eyes with active nAMD despite the consequent anti-VEGF treatment every 4–8 weeks, indicating its superior efficacy regarding anatomic criteria [5,6,7]. Adding to this short-term experience, the aim of this study was to evaluate the functional and anatomic outcomes after switch to brolucizumab over 6 months in eyes with active nAMD under treatment intervals of ≤6 weeks with other anti-VEGF agents.

## 2. Materials and Methods

This is an ongoing prospective open-label single-center (Berner Augenklinik am Lindenhofspital) cohort study, analyzing the effect of a switch to brolucizumab in eyes with nAMD insufficiently responsive to ranibizumab or aflibercept. Primary endpoints are the change in morphologic markers of disease activity measured by central subfield thickness (CST) and the presence of intra- and subretinal fluid, as well as single optotype distance and text reading visual acuity. Secondary outcomes include changes in injection intervals, annual number of injections, and qualitative changes in activity signs of macular neovascularization (MNV) using optical coherence tomography angiography (OCTA). The study is designed for 2 years. Here, we report the preliminary 6 month data.

The study followed the standards of the Declarations of Helsinki, and local ethical board approval was granted (Kantonale Ethikkommission Bern, Switzerland, reference number 2020-00412). Patients from February to June 2020 were offered a switch to brolucizumab with the following inclusion criteria: treatment for nAMD for at least 1 year and functionally relevant persisting intra- and/or subretinal fluid despite treatment intervals of 6 weeks or less. Exclusion criteria were denial of informed consent, macular scarring preventing a change in visual function, and other causes of intra- or subretinal fluid. Treatment-naïve patients were not included in this study. Treatment was switched after the confirmation of MNV activity by dye leakage in fluorescein angiography (FA) and persisting fluid in optical coherence tomography (OCT). Initial therapy was initiated with three monthly loading injections using ranibizumab or aflibercept at the discretion of the treating physician, thereafter following a treat and extend (T&E) regimen with 2 weeks increments according to morphological signs of activity in OCT. Upon switching, treatment with brolucizumab was initiated within maximally 6 weeks of the last injection, followed by a second injection after 1 month, thereafter again following a treat-and-extend protocol (T&E) according to morphological response in OCT. Treatment intervals were extended by 2 weeks in the case of resolved intra- and resolved or stable subretinal fluid over three consecutive injection intervals. At every visit, best corrected distance visual acuity (BCVA) was measured on a Snellen decimal scale and converted to Early Treatment of Diabetic Retinopathy Study (ETDRS) letters equivalent, where a Snellen decimal BCVA of 1.0 was defined as 85 ETDRS letters for the statistical analysis; reading acuity (RA) was measured on a standardized text, which is comparable to Radner and Birkhäuser charts, with logarithmically reduced font size, ranging from 0.1 to 1.6 Snellen decimal equivalent (https://szb.abacuscity.ch/de/A~51.952/Nahsehprobe-D-Erw.-R%C3%BCckseite%3A-ETDRS-_-LCS, accessed on 1 June 2021). Results were converted to the negative logarithm of decimal reading acuity (logReading Acuity Determination = LogRAD) for statistical analysis. OCT angiography (OCTA) was also carried out (Heidelberg Spectralis OCT 2 with 880 nm wavelength, axial resolution 3.9 µm and lateral resolution 5.7 µm, Heidelberg Engineering, Heidelberg, Germany). Lastly, a thorough ophthalmologic examination including dilated fundoscopy was performed to exclude signs of inflammation. Central subfield thickness was defined as the average thickness between the internal limiting membrane (ILM) and Bruch’s basal membrane (BM) within the central 1 mm of the fovea. Centering to the fovea and BM segmentation were manually controlled and adjusted if needed for fixation or segmentation misalignments. The 6 month data were analyzed using SPSS (software package V.23 (IBM Inc., Armonk, New York, NY, USA)). Changes of BCVA in ETDRS letters, RA in LogRAD, CST, injection intervals, and number of injections are reported with their respective 95% confidence intervals (CIs). Given the small sample sizes, we decided not to apply statistical analyses for this descriptive dataset.

## 3. Results

Twelve eyes of 12 patients were switched to brolucizumab according to the abovementioned criteria between February and May 2020 in our center. The complete baseline characteristics and principal measurements are listed in Table 1.

Seven eyes reached a follow-up of 6 months (28.0, SD ± 2.8 weeks), in which they received 4.4 ± 0.5 brolucizumab injections compared to 4.7 ± 0.8 during the 26 weeks before baseline. The mean treatment interval was extended from 5.3 ± 0.9 weeks before switch to brolucizumab 9.0 ± 2.8 weeks thereafter (Figure 1, 95% CI of extension: 1.6 to 5.9 weeks).

Central subfield thickness regressed from 422.1 ± 97.3 µm to 353.6 ± 100.9 µm (95% CI: −19.9 to 157.1). A complete resolution of the previously persisting intraretinal and/or subretinal fluid was found in two out of seven eyes (29%). Intraretinal fluid was present in four (57%) and subretinal fluid in three eyes (43%) at baseline, and intraretinal fluid was present but reduced in two (29%, example in Figure 2) and subretinal fluid remained present in three eyes (43%), to a lesser extent in two eyes, and increased in one eye after 6 months. Distance BCVA changed from 67.8 ± 7.2 at baseline to 72.2 ± 7.5 ETDRS equivalents after 6 months (Figure 3a, 95% CI: −0.3 to 9.1). Reading acuity improved from 0.48 ± 0.15 at baseline to 0.31 ± 0.17 LogRAD after 6 months (Figure 3b, 95% CI: 0.03 to 0.25), which represents a two-line increase in RA from approximately 0.32 to 0.5 Snellen decimal. Table A1 (Appendix A) shows all measurements per patient at baseline and at the 6 month follow-up visit.

Five patients were excluded from the analysis because treatment had to be discontinued prior to reaching the prescheduled study end after 6 months. Two patients developed intraocular inflammation with extramacular vascular occlusion after one and two injections, respectively, needing systemic corticosteroid treatment and the discontinuation of brolucizumab therapy, followed by a switch back to the previous anti-VEGF agents. The two cases are described in detail in the next section. One patient suffered a transitory ischemic attack. Since he had a previous history of stroke, this was probably not attributed to brolucizumab, but the therapy of the affected poorer eye was discontinued as a precautionary measure, and later resumed with the previous anti-VEGF agent. One patient with massive intraretinal fluid subjectively experienced further visual deterioration despite better visual acuity after one brolucizumab injection and requested to be switched back to the previous anti-VEGF therapy. In one patient, therapy was supplemented with intravitreal dexamethasone to reduce persistent injection burden by the attending physician, according to reported experience [8]. In the analysis of the excluded subset, no statistically significant change in BCVA or CST after 6 months compared to baseline was observed (Table A1, Appendix A).

### Case Reports: Two Patients with Intraocular Inflammation after Brolucizumab

Case 1: 86 year old woman with a history of myocardial infarction. After 22 anti-VEGF injections within 27 months, treatment was switched to brolucizumab. Four weeks after switching to brolucizumab and escaping the patient’s attention, a two-line reduction in best corrected visual acuity (BCVA) from 20/32 to 20/50, confluent keratic precipitates (Figure 4A), and moderate anterior and intermediate uveitis (2+ anterior chamber and vitreous cells, vitreous haze 2+) were observed. Wide-field fluorescein angiography showed arterial branch occlusions in the inferior vessel arcades (Figure 4B) without vascular leakage. Three-hourly treatment with topical prednisolone and systemic prednisolone 1 mg per kg of body weight (mg/kg of body weight per day for 1 week was tapered off thereafter while continuing acetylsalicylate and clopidogrel. After 1 week, precipitates disappeared. Ocular inflammation improved within 2 weeks, while BCVA decreased to 20/63 before recovering to 20/40 after 1 month. Intraretinal fluid was reduced for 2 months before increasing to the baseline-level 12 weeks after brolucizumab injection, at which time treatment with aflibercept was resumed.

Case 2, 84 year old male with a history of myocardial infarction, stroke, and coronary and valvular surgery, under hemodialysis for renal failure after glomerulonephritis. After a total of 41 intravitreal injections within 72 months and significant retinal fluid despite a treatment interval of 4–5 weeks in his better eye, treatment was switched to brolucizumab. Four weeks after the second brolucizumab injection, the patient reported floaters, whereas BCVA improved from 20/40 to 20/32 and IRF resolved. We found a significant panuveitis with preretinal infiltrates, but no vascular sheathing. Widefield angiography revealed segment arterial leakage and extramacular branch arterial occlusions (Figure 5A–D). Intravenous methylprednisolone (40 mg) was given on two consecutive days, followed by oral prednisolone for 3 days (at 1 mg/kg bw then tapered off over 1 month), accompanied by acetylsalicylate. After 1 month, BCVA remained 20/32 in the absence of IRF with persisting vitreal infiltration. Further follow-up showed stable BCVA, recurrence of IRF, and retrograde staining of the occluded vessels 10 weeks after the last brolucizumab injection.

## 4. Discussion

Our data indicate a benefit of switching to brolucizumab with an increased reading visual acuity and increased treatment interval in eyes that were insufficiently responding to previous anti-VEGF agents in intervals of ≤6 weeks following a T&E regimen for nAMD. This confirms previous reports of a reduction in CST within the first 2 months after the switch to brolucizumab [5,6,7], and better morphological outcomes and a functional noninferiority compared to aflibercept over 2 years in two phase 3 studies [3]. Moreover, in accordance with previous studies, distance BCVA did not change significantly after switching from other anti-VEGF agents to brolucizumab, which is well in line with previous experiences of switching from ranibizumab to aflibercept [8,9,10,11,12,13,14,15]. In contrast to these publications, a sustained reduction in CST after 6 months was not recorded in our cohort, which may be linked to insufficient power. Nevertheless, we found a sustained improvement of reading acuity after 6 months. Along with a high satisfaction of our successfully switched patients due to a perceived improvement in visual performance, this provides further support that single-letter distance visual acuity is not representative of daily visual tasks [12,16,17,18].

A correlation of retinal thickness in OCT with reading abilities has been demonstrated in nAMD [19], but reading tests are not widely adapted in clinical practice and research, and testing is not standardized [20]. While ETDRS reading scores are standard for functional outcomes in large randomized studies, vision-related quality of life correlates more with contrast sensitivity and binocular reading speed, but ETDRS letter score does not correlate with binocular reading speed [16]. Reading speed and comprehension are substantially reduced in AMD [21], and reading disabilities have a significant impact on quality of life in AMD [22]. Furthermore, microperimetry revealed correlations of reading speed with fixation stability, foveal absolute scotoma, reading acuity, and sociodemographic characteristics [23]. Similar findings are known from postoperative outcomes after the peeling of epiretinal membranes, where an increase in reading visual acuity and critical print-size was shown in patients despite unchanged BCVA [24]. Taken together, visual acuity measured with single optotypes seems unable to represent pathological changes in AMD or relevant parameters for the quality of life of affected patients. Reading requires the recognition of closely spaced letters or signs, which requires a larger parafoveal area of the retina than simply the resolution between two points. The discrepancy of impaired reading with maintained visual acuity is explained in diseases such as nAMD by the many small microlesions of the sensory cells in the center of the macula. To address this problem and achieve a more patient-centered therapy, different visual assessments have been compared [20]. Our data support the inclusion of reading acuity in the routine assessments of patients with nAMD as it has a greater informative value with regard to vision-related quality of life, as well as pathophysiology.

An obvious limitation of our study is the small sample size and the large dropout rate, together with a relatively short follow-up period. With respect to this, we decided to report only 95% confidence intervals without formally stating significance. The high dropout rate is explained by a surprisingly high incidence of treatment complications in our cohort with two cases of relevant intraocular inflammation and retinal vascular occlusion after one and two brolucizumab injections. HAWK and HARRIER already reported relatively high rates of intraocular inflammation (IOI) in eyes treated with brolucizumab but a similar rate of severe vision loss compared to the controls treated with aflibercept [3]. The HAWK and HARRIER reported rates for Aflibercept were, however, also higher than observed in a review of 10 phase 3 trials for aflibercept [25]. After the regulatory approval of brolucizumab, a case series of intraocular inflammation with retinal vascular occlusion was reported, some of them with significant vision loss [26,27,28]. As a result of these reports, treatment preferences are based on expert opinions, and the impact of treatment choice on visual outcome in real life has remained narrative [29,30]. The pathophysiological basis and risk factors for this IOI signal are currently a topic of intensive research within the community and the manufacturer. A post hoc analysis of the study data by an expert committee showed that most IOI with severe visual loss happened within the first 3–6 months of treatment [31]. After analysis of first interpretable data of the MERLIN trial, on 28 May 2021, Novartis^®^ advised against the use of brolucizumab with intervals shorter than 6 weeks after up to three monthly loading injections, and terminated ongoing studies that allowed this possibility [32]. Therefore, efficacy and possible therapeutic improvement must be balanced against a potential risk, while our cohort showed a disproportionate incidence of such responses. This said, both of our patients recovered above baseline after treatment with topical and systemic corticosteroids, before treatment was resumed with the previous anti-VEGF agents.

In summary, in this small cohort, we observed an improved reading acuity and longer treatment intervals 6 months after switch to brolucizumab in patients with persistent nAMD activity despite intensive anti-VEGF treatment. Larger sample sizes are required to estimate the strength of this effect, which will prospectively be addressed in the currently recruiting Falcon study (ClinicalTrials.gov Identifier: NCT04679935).

## 5. Conclusions

Based on the preliminary data of our cohort, brolucizumab led to an increased reading acuity and longer treatment intervals in patients with high treatment demand for other anti-VEGF drugs, while distance BCVA measured with standard ETDRS or Snellen charts and CST remained unchanged. It, thus, seems that brolucizumab has a place at least as a second-line anti-VEGF agent in patients with high treatment demand. The increase in reading acuity seems to go along with an improved control of disease activity after switch to brolucizumab and may well contribute to an improved vision-related quality of life. Given the risk of intraocular inflammation and vascular occlusion, careful patient selection and education remain essential for early detection and successful treatment of possible complications. We strongly advocate the use of standardized reading charts for the assessment of functional evolution in nAMD under treatment, appearing to be a better physiological and psychological representation of the functional impact of disease activity with regard to vison-related quality of life.

## Figures and Tables

**Figure 1 jcm-10-02666-f001:**
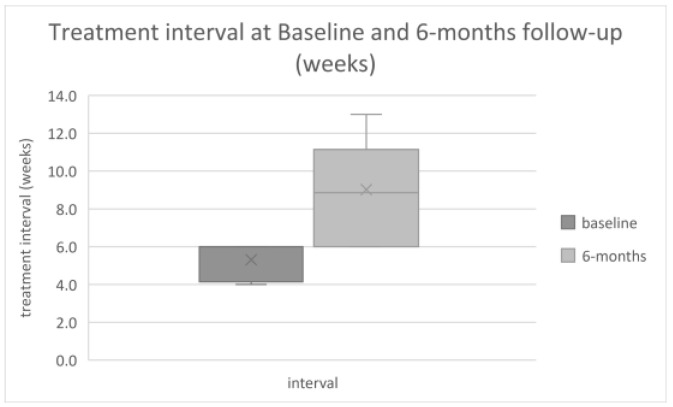
The last treatment interval at baseline before switch to brolucizumab was 5.30 ± 0.93 weeks with persistent disease activity before switch to brolucizumab and 9.0 ± 2.8 weeks at 6 months (95% CI of extension: 1.6 to 5.9, *n* = 7).

**Figure 2 jcm-10-02666-f002:**
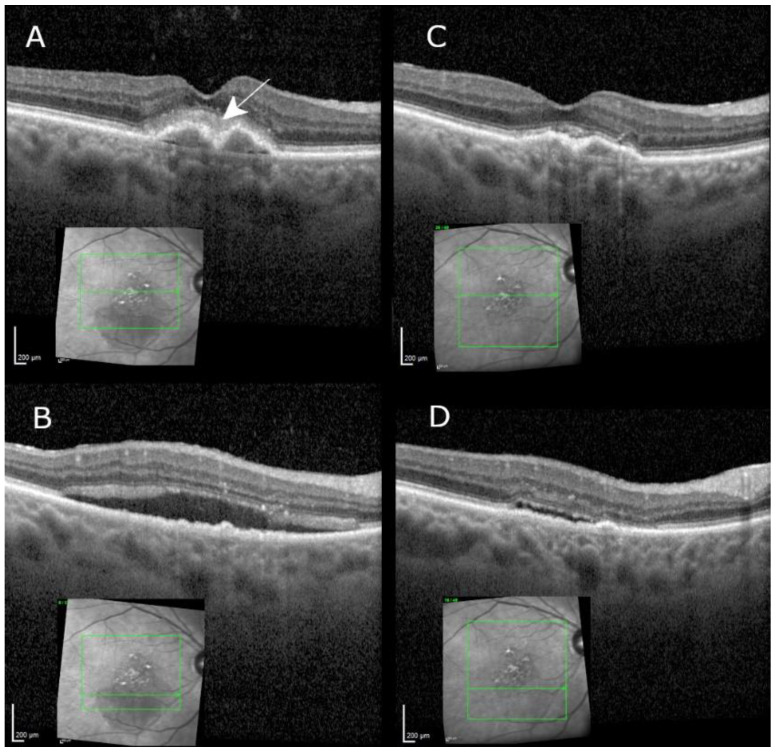
Optical coherence tomography (OCT) images of an 85 year old patient through the fovea (upper row, (**A**,**C**)) and the inferior parafoveal macular area (lower row (**B**,**D**)) at switch, 6 weeks after the last anti-VEGF injection (left row (**A**,**B**)), at the 6 month measurement visit 29 weeks from baseline, and 11 weeks after the fourth Brolucizumab injection. The figure shows resolved foveal subretinal hyperreflective material (arrows) and massively regressed subretinal fluid inferior to the fovea, despite a longer treatment interval. Meanwhile, distance visual acuity improved from 60 to 75 ETDRS letters, and reading acuity improved from 0.25 to 0.32 Snellen decimal (0.6 to 0.5 logRAD = logReading Acuity Determination).

**Figure 3 jcm-10-02666-f003:**
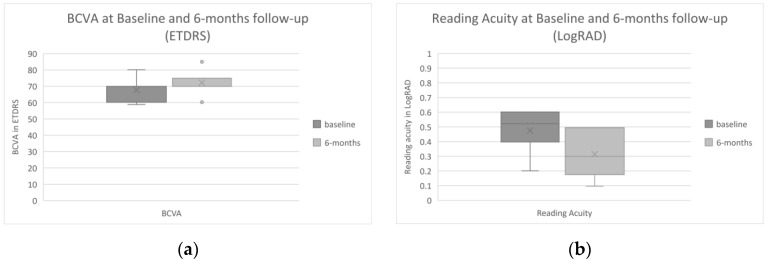
Box-and-whisker graph indicating changes in best corrected visual acuity (BCVA, 1A) and reading acuity (RA, 1B) from baseline to the 6 month follow-up exam. BCVA (**a**) was 67.8 ± 7.2 at baseline and 72.2 ± 7.5 ETDRS equivalents after 6 months (95% CI: −0.3 to 9.1, *n* = 7). Reading acuity (**b**) improved from 0.48 ± 0.15 (approximately 0.32 Snellen decimal) at baseline to 0.31 ± 0.17 (approximately 0.5 Snellen decimal) LogRAD after 6 months (95% CI: 0.03 to 0.25, *n* = 7), indicating a possible two-line increase in reading acuity.

**Figure 4 jcm-10-02666-f004:**
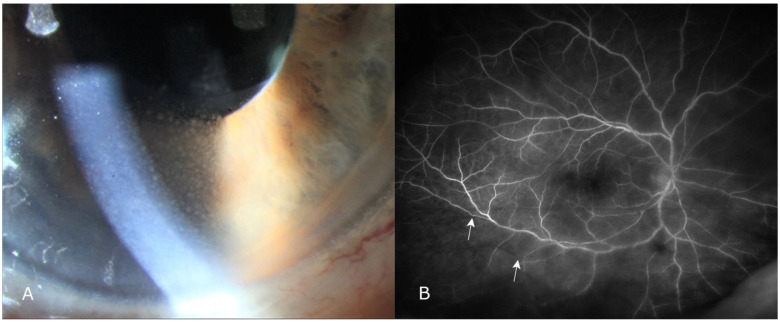
Case 1, an 86 year old woman with anterior uveitis and confluent mid-sized keratic precipitates (**A**): slit-lamp photograph and intermediate uveitis with mid-peripheral arterial occlusion of two arterial branches (arrows, (**B**): wide-field fluorescein angiography) 4 weeks after the first intravitreal brolucizumab injection. Intraretinal fluid slightly regressed for 12 weeks, and anterior uveitis and keratic precipitates completely regressed within 2 weeks of systemic and topic prednisolone. Visual acuity regained close to baseline values after temporary deterioration.

**Figure 5 jcm-10-02666-f005:**
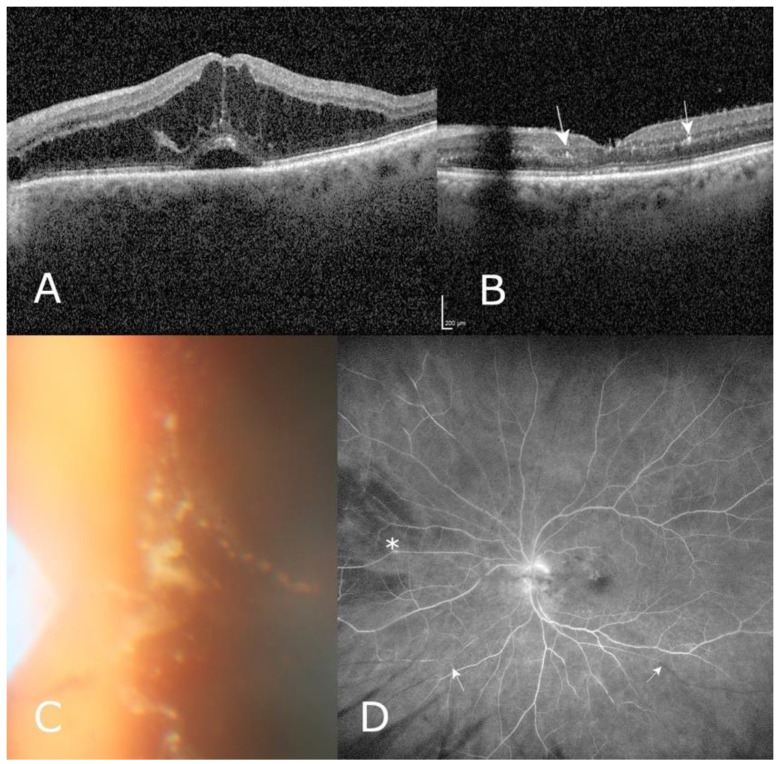
Case 2, an 84 year old male with persistent significant intraretinal fluid in optical coherence tomography (**A**) despite monthly treatment with aflibercept for age-related macular degeneration. Six weeks after the second four-weekly brolucizumab injection, intraretinal fluid completely regressed (**B**). The patient reported floaters without visual deterioration, and posterior uveitis with typical preretinal infiltrates (arrows indicate intraretinal hyperreflective spots in (**B**)) was observed (**B**,**C**). Wide-field fluorescein angiography (**D**) revealed vasculitis (asterisk) and branch retinal artery occlusion (arrows). Four weeks later, fluorescein angiography (not shown) revealed a probably retrograde reperfusion of previously occluded arterial branches.

**Table 1 jcm-10-02666-t001:** Baseline characteristics of patients and reason for dropouts. Legend: #IVT, total number of IVTs applied in total (months since baseline and number of injections) and within the last 6 months (number); m, male; f, female; R, right; L, left.

	Demographic and Disease Characteristics						Treatment Discontinuation
Measure	Age	Sex	Laterality	#IVT Pretreatment			
				Total		6 Months	
Unit	Years	m/f	R/L	Months	*n*	*n*	
Included	77	m	R	61	51	4	no
	84	f	R	27	19	4	no
	65.1	f	L	75	42	5	no
	77.2	f	L	113	91	6	no
	85.1	f	R	34	24	4	no
	84.6	m	R	53	30	5	no
	76.1	m	L	54	46	5	no
Excluded	76.3	m	R	13	11	5	14 weeks from baseline: transitory ischemic attack, 3 weeks after third brolucizumab injection. Not related to treatment. Discontinuation for safety precautions.
	84.4	f	R	27	22	6	4 weeks after baseline: anterior and intermediate uveitis without vasculitis, 4 weeks after first brolucizumab injection.
	78.1	m	L	72	41	5	10 weeks after baseline: panuveitis and retinal occlusive vasculitis 6 weeks after second brolucizumab injection.
	87.1	f	L	13	14	6	4 weeks after baseline: progressive pigment epithelial detachment; switched back to aflibercept upon decision of the treating physician.
	88.8	f	R	48	10	5	21 weeks after baseline: adjunctive treatment with intravitreal dexamethasone for persistent intravitreal fluid, 5 weeks after the fifth brolucizumab injection (see text).

## Data Availability

The data presented in this study are available in Table A1 (Appendix A), and further details are available on request from the corresponding author. More detailed data are not publicly available due to local data privacy regulations.

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
