# Peer review of "Switching to Brolucizumab in Neovascular Age-Related Macular Degeneration Incompletely Responsive to Ranibizumab or Aflibercept: Real-Life 6 Month Outcomes"

_jcm, 2021, doi:10.3390/jcm10122666_

Round 1
Reviewer 1 Report
In their manuscript, the authors describe the response of patients switched to brolicuzimab after treatment with other VEGF antagonists. The study is comprised of a very low sample size and merely descriptive, but may of interest. There are some aspects that need to be considered before publication can be considered. Especially, more data need to be provided.
Abstract
The surprisingly high incidence of inflammation after the treatment with brolicuzimab (2 out of 12 patients) is of concern. The authors need to include this into the conclusion of the abstract.
Results
I am not sure if I understood correctly, the number of injections developed from 4.3 to 4.9? This does not sound like a reduction of injections. This should be covered in the discussion.
The authors are very scarce with the presentation of the patients. Please include a table of the baseline information, (age, gender, BVCA, retinal thickness, number of injections, duration of disease etc.) of the patients. Please provide OCT and fundus of patient examples with and without brolicuzumab. Please include data/fundus pictures etc. on the SAE patients.
Discussion
The disturbing incidences of intraocular inflammation after brolicuzimab treatment need to mentioned in the summary of the discussion. This is an important finding.
Reviewer 2 Report
This study showed outcomes of switching to brolucizumab. This is one of the hot current topics in the field of retina. The study is generally well-designed and shows important information. However, I have several concerns.
- Detailed characteristics of the patients, such as age, sex, duration between the initial diagnosis of neovascular AMD and the switching, type and number of anti-VEGF injections administered before switching to brolucizumab, BCVA before and after the switching, and so on… A table showing characteristics of each patient is needed.
- Please add a figure showing change in OCT features before and after brolucizumab injection.
- If possible, can you present the change in choroidal thickness before and after the switching to brolucizumab?
- Intraocular inflammation after brolucizumab injection is one of the hot current issues. Please make a brief description about the characteristics of two patients developed intraocular inflammation: age, sex, type and number of anti-VEGF injections administered before switching to brolucizumab
- Measuring reading acuity is the strong point of this study. In this study, the mean reading acuity was improved from 0.48 to 0.31. Please provide a short discussion about the clinical importance of this improvement.
Round 2
Reviewer 1 Report
I would suggest acceptance, as the paper has been significantly improved. by the authors.Reviewer 2 Report
All the queries were well-addressed.
I have no more comment.